# Preclinical and Basic Research Strategies for Overcoming Resistance to Targeted Therapies in HER2-Positive Breast Cancer

**DOI:** 10.3390/cancers15092568

**Published:** 2023-04-30

**Authors:** Yi Cao, Yunjin Li, Ruijie Liu, Jianhua Zhou, Kuansong Wang

**Affiliations:** 1Department of Pathology, Xiangya Hospital, Central South University, Changsha 410008, China; 2Department of Pathology, School of Basic Medical science, Central South University, Changsha 410008, China

**Keywords:** HER2-positive breast cancer, targeted therapy, resistance, anti-resistance

## Abstract

**Simple Summary:**

The development of drug resistance in HER2 targeted therapies poses a major challenge in breast cancer treatment. To address this challenge, researchers have adopted two main strategies: repurposing existing drugs to enhance their effectiveness and identifying new targets to broaden the range of therapeutic options. This review discusses current research efforts in these two areas, highlighting key points and challenges. By pursuing these strategies, researchers hope to improve treatment outcomes and overcome drug resistance, ultimately leading to better outcomes for patients with HER2-positive breast cancer.

**Abstract:**

The amplification of epidermal growth factor receptor 2 (HER2) is associated with a poor prognosis and *HER2* gene is overexpressed in approximately 15–30% of breast cancers. In HER2-positive breast cancer patients, HER2-targeted therapies improved clinical outcomes and survival rates. However, drug resistance to anti-HER2 drugs is almost unavoidable, leaving some patients with an unmet need for better prognoses. Therefore, exploring strategies to delay or revert drug resistance is urgent. In recent years, new targets and regimens have emerged continuously. This review discusses the fundamental mechanisms of drug resistance in the targeted therapies of HER2-positive breast cancer and summarizes recent research progress in this field, including preclinical and basic research studies.

## 1. Introduction

HER2 (also known as Neu or ErbB2) is a member of the epidermal growth factor receptor (EGFR; also known as ErbB) family of receptor tyrosine kinases (RTK), which includes HER1(EGFR, ERbB1), HER2, HER3 (ERbB3), and HER4 (ERbB4) in human [1]. The ErbB network can be divided into three parts: the input part (Figure 1a), which consists of ligands (EGFs) and their four receptors (EGFRs), receives the signal and transmits the information to the signal processing part (Figure 1b), including but not limited to the MAPK, JAK-STAT3, and PI3K/AKT/mTOR pathways. Finally, the liner cascades of these pathways activate the output part, resulting in cell division and migration (both associated with tumorigenesis), adhesion, differentiation, and apoptosis [2] (Figure 1c). The overexpression of HER2 will cause more heterodimer formation, and the HER2-containing heterodimer is the most transforming and mitogenic receptor complex [2]. Consequently, HER2 overexpression has close relation with a poorer prognosis, which is detected in 15–30% of all invasive breast cancer (BC) patients [3]. Notably, an advanced subtype of BC has emerged: HER2-low BC, which is defined as HER2 immunohisto chemistry IHC 1+ or IHC 2+ and in situ hybridization (ISH)-negative [4]. In this type of BC, targeted therapies are also effective and beneficial to patients’ survival. There are several clinical trials that have enrolled patients with HER2-positive and HER2-low expression breast cancer. We have summarized these trials in Table 1 to provide an intuitive comparison between these two populations, who were treated with the same agents but had different outcomes [5,6,7,8,9,10].

Before the advent of HER2-targeting therapies, the prognosis of HER2-positive BC was poor due to fast tumor growth and impaired response to chemotherapy. The introduction of anti-HER2 therapies has dramatically improved clinical outcomes. Despite these advances, the drug resistance continues to pose a significant challenge in the application of anti-HER2 therapies. Overcoming drug resistance is an urgent matter. HER2-positive breast cancer patients who develop resistance to targeted therapies may experience tumor recurrence or progression, reduced treatment efficacy, and increased adverse effects. The American Society of Clinical Oncology (ASCO) Guideline [11], updated in 2022, now presents existing therapy regimens to address this issue [11], which was updated in 2022 (Figure 2).The mechanisms of drug resistance are complicated, such as lower binding between agents with HER2 [12], ErbB2 mutation [13,14], epigenetic changes [15], alteration of the tumor microenvironment [16], tumor stem cell self-renewal [17], and the overactivation of the signaling pathways [18]. There has been increasing research efforts in finding the solutions to drug resistance, which primarily focused on enhancing the effectiveness of existing HER2-targeted agents approved by the Food and Drug Administration (FDA) or developing novel anti-resistance agents. In this review, we briefly summarized new understandings of the drug resistance mechanisms and potential strategies to overcome resistance to HER2-targeted therapies in recent years. Our review outlines a brief and precise framework of preclinical and basic research strategies for overcoming resistance to targeted therapies in HER2-positive BC.

## 2. Investigating Regimens for Overcoming Resistance: Insights from Preclinical Studies

Currently, targeted therapies for HER2 have significantly improved the survival rate of HER2-positive BC patients. So far, the FDA has approved several targeted therapies to treat HER2-positive BC, including monoclonal antibodies (mAb), such as trastuzumab, pertuzumab, tyrosine kinase inhibitors (lapatinib, neratinib, tucatinib), and antibody-drug conjugates (DS-8201a, T-DM1, TDxd). Trastuzumab entered clinical trials in the early 1990s. As a recombinant humanized monoclonal antibody, it combined with the extracellular domain of cell-surface receptor HER2 [19] disturbed the dimerization of HER2, and blocked the downstream signaling, leading to decreased proliferation of tumor cells. Meanwhile, the working mechanism of pertuzumab was similar to trastuzumab. However, pertuzumab exerted a stronger effect than trastuzumab [20]. As its name implies, tyrosine kinase inhibitor (TKI) blocked signal transduction, inhibiting malignancy development. There are some differentiations among these agents. For example, lapatinib is EGFR/HER2 TKI [21], neratinib is a pan-HER TKI [22], and tucatinib is the selective HER2-targeted TKI [23].

Nevertheless, the resistance of HER2-targeted therapies always leads to adverse effects on patients’ outcomes. Several resistance mechanisms of the monoclonal antibody trastuzumab (mAb) have been illustrated, including impaired binding to the HER2 receptor due to HER2 levels and variants (with the p95HER2 isoform being the most studied), as well as molecular masking [12]. Additionally, aberrant activation of the PI3K/AKT/mTOR signaling cascade can contribute to mAb resistance [24], since the inhibition of this cascade by mAb is crucial for its anticancer effects. Impaired antibody-dependent cellular cytotoxicity (ADCC), which involves the recruitment and activation of immune cells to kill cancer cells, can also contribute to mAb resistance [25].

Similarly, hyperactivation of the PI3K/AKT/mTOR and MAPK pathways contributes to TKI resistance. Notably, the HER2 variant p95HER2 can induce resistance to mAb but remains sensitive to TKIs. However, *HER2* gene mutation can impair the binding between TKIs and HER2 [26]. The activation of pathways parallel to HER2 also contributes to TKI resistance.

Currently, there is a consensus on the resistance mechanisms of antibody-drug conjugates (ADCs). Downregulation and/or mutation of the targeted cell surface antigen can cause loss of antibody-mediated activity [27]. Additionally, impaired internalization and trafficking pathways of ADCs and excessive efflux from targeted cells contribute to resistance [28].

To overcome this obstacle, researchers have made a great effort to explore potential strategies, such as more effective synergistic combinations and identification of new targets. In this part, we overviewed the progress of the FDA-approved HER2-targeted agents published in recent years. Most of the patients in these trials have metastatic breast cancer (MBC) or previously received targeted therapies, which indicated potential drug resistance in the population (Table 2).

### 2.1. Monoclonal Antibody Trastuzumab

In the phase II eribulin–HP (trastuzumab–pertuzumab) trial (ClinicalTrials.gov identifier NCT01912963; *n* = 32), patients with metastatic, unresectable, locally advanced, or locally recurrent HER2-positive breast cancer, after the run-in period, the study expanded to two cohorts: Cohort A (*n* = 19) included patients who had no prior pertuzumab exposure, while Cohort B (*n* = 6) included patients who had received pertuzumab. After up to six cycles (cycle duration = 21 days) of combination therapy, the result indicated that in the HER2-positive patients without prior pertuzumab exposure, eribulin–HP had manageable toxicity and modest clinical activity. Additionally, genetic analysis in patients with no clinical response to eribulin–HP observed a higher TP53 mutation, supporting that TP53 mutation is associated with the worsened treatment response and resistance. However, further investigation is needed to validate this potential role of TP53 alterations in resistance to treatments for HER2-positive BC [29].

The PEONY phase III randomized clinical trial (ClinicalTrials.gov identifier NCT02586025) carried on 329 patients with early stage or locally advanced HER2-positive breast cancer in an Asian population. Patients were randomized 2:1 to receive four cycles (cycle duration = 21 days) of intravenous pertuzumab, trastuzumab, and docetaxel or intravenous placebo, trastuzumab, and docetaxel before surgery. After surgery, the patients received the same regimens for 13 cycles, along with three cycles of intravenous fluorouracil, epirubicin, and cyclophosphamide. The trial concluded that the addition of pertuzumab to trastuzumab and docetaxel led to a statistically significant improvement in the total pathologic complete response (tpCR) rate and was found to have a safety profile in line with the known pertuzumab safety profile. However, the secondary endpoints (event-free survival and overall survival) could not be thoroughly evaluated due to the short follow-up period at the clinical cutoff date [30].

The above two trials pointed to the combination of mAb. In contrast, PATRICIA (ClinicalTrials.gov identifier: NCT02536339) studied from another angle. It was a phase II, interventional, open-label, and single-arm trial that investigated the dose-dependent response (6 mg/kg intravenous once weekly) to trastuzumab in HER2-positive MBC with central neural system (CNS) metastases and CNS progression. The results indicated that dose alteration of mAb might have clinical utility and support the feasibility of optimizing the dose and schedule of mAb to combat CNS metastases [31].

### 2.2. Tyrosine Kinase Inhibitors

An international, randomized, double-blind trial (ClinicalTrials.gov identifier: NCT02614794) was conducted to determine whether the combination of tucatinib plus trastuzumab and capecitabine was superior to placebo plus trastuzumab and capecitabine. The trial assigned 612 patients with HER2-positive MBC into 2:1 for two groups. After 1 year of different treatments, the group added a proven active combination tucatinib, which resulted in better progression-free survival and overall survival outcomes [23]. Meanwhile, the patient’s life quality was preserved [37]. Notably, the enrolled patients included a large percentage of patients with previously treated progressing brain metastases (BM), and tucatinib could pass through the blood–brain barrier, making it an effective option for combating brain metastases. As a result of its reliable outcomes, tucatinib received FDA approval in April 2020.

A phase II trial NCT00684983 was conducted to investigate the hypothesis that adding cituxumumab, an inhibitor of the insulin growth factor-1 receptor (IGF-1R), to the regimen of lapatinib and capecitabine would lead to improved prognosis and overcome trastuzumab resistance in patients with HER2-positive advanced BC. However, there were no significant results on whether median progression-free survival (mPFS), median overall survival (mOS), or quality of life (QoL) between groups in this trial [32]. Although the results failed to meet the expected assumption, it did not mean the destination of IGF-R1 in HER2-positive BC, some research has found another promising perspective, and we discussed it in Section 2.1.

A randomized, active-controlled phase III NALA trial (ClinicalTrials.gov identifier NCT01808573) enrolled 621 patients who were HER2-positive MBC and received ≥ 2 prior lines of HER2-directed therapy previously. These patients were assigned 1:1 into N + C (neratinib + capecitabine) and L + C (lapatinib + capecitabine). Meanwhile, patients in N + C took antidiarrheal prophylaxis to prevent gastrointestinal toxicity. The results suggested that N + C compared to L + C, could significantly improve PFS and delay the development of CNS disease. No new N + C safety signals were observed [33]. Like tucatinib, combined neratinib and capecitabine had been approved by FDA to treat HER2-positive MBC patients who have received ≥ 2 prior HER2-directed regimens in February 2020.

### 2.3. Antibody-Drug Conjugates

The trial KATHERINE showed the superiority of trastuzumab emtansine (T-DM1) in the patients with the residual invasive disease who had received a taxane-trastuzumab-containing regimen [38]. In this context, in the KAITLIN study (NCT01966471), 1846 postoperative patients with HER2-positive early breast cancer were randomized into two groups. The first group received three to four cycles of anthracycline-based chemotherapy (AC) followed by 18 cycles of T-DM1 plus pertuzumab (AC-KP), while the second group received three to four cycles of AC followed by 18 cycles of trastuzumab and pertuzumab (AC-THP). However, in both node-positive subpopulation and intention-to-treat overall population, treatment with AC-KP failed to reduce the risk significantly of an invasive disease-free survival (IDFS) event compared with AC-THP, when two arms had similar rates of grade ≥ 3 and serious adverse events (AEs) [34]. The above evidence indicates that AC-THP remains important. Notably, compared with THP, KP decreased clinically meaningful deterioration (HR, 0.71; 95% CI, 0.62 to 0.80) in overall health status. Similarly, another research (NCT01702558) compared the efficacy and safety of the regimens of T-DM1 plus capecitabine and T-DM1 alone and showed no significant difference between these two groups [35].

To evaluate the effectiveness of T-DM1 in the treatment of recurrent or metastatic HER2-positive breast cancer, the phase IIIb clinical trial, also known as KAMILLA (ClinicalTrials.gov identifier NCT01702571), was conducted in a cohort of 2185 patients with HER2-positive metastatic breast cancer, including those with brain metastases. The entire population was assigned 1:4 into BM and without BM. All the participants received T-DM1 3.6 mg/kg intravenously on Day 1 of a cycle (cycle duration = 21 days) every three weeks until unacceptable toxicity, withdrawal of consent, or disease progression. The results indicated that T-DM1 was effective and well-tolerated in patients with baseline brain metastases, and potentially beneficial even after local treatment [36]. However, patients in KAMILLA did not receive pertuzumab treatment which is the current first-line treatment of MBC. There exists controversy that in patients previously exposed to pertuzumab, the efficacy of T-DM1 could be reduced potentially. Furthermore, the team confirmed that RAB5A protein might be a promising biomarker to predict the sensitivity of T-DM1, which was verified it in selected patients in the KAMILLA study [39].

The phase 2 ATEMPT trial (ClinicalTrials.gov identifier NCT01853748) was carried out to determine whether T-DM1 is superior to TH regimens (paclitaxel plus trastuzumab) in the safety assessment. The trial randomized 512 patients 3:1 to receive T-DM1 for a total of 51 weeks, or TH for 12 weeks, followed by trastuzumab every three weeks for 13 cycles. Although the results indicated an insignificant difference of clinically relevant toxicities (CRTs), which was the co-primary endpoint of the trial, there were low rates of AEs occurred and improved QoL in T-DM1 treatment. The 3-year DFS was 97.7% in the T-DM1 arm, comparing with 92.8% in the TH arm [40]. Given the above evidence, T-DM1 could be an alternative plan when stage I HER2-positive BC had TH-related side effects.

## 3. Investigating Novel Strategies for Overcoming Resistance: Insights from Basic Studies

The aforementioned studies have primarily focused on the clinical exploration of targeted drugs that have been utilized in clinical settings. While some studies have yielded anticipated positive results, others have presented negative results. Nevertheless, these results hold valuable and significant guidance for future research endeavors. In addition to exploring the repurposing potential of existing drugs, researchers are increasingly concentrating on discovering novel potential targets or resistance mechanisms. However, most of these studies are still in the basic research phase. There is a significant journey ahead before their translation into clinical settings. We summarize and categorize these basic research studies into three parts based on the ErbB network to facilitate a better understanding of recent developments in this field. As described above, the ErbB network can be divided into the input, signal processing, and output parts.

### 3.1. The “Input Part” of the ErbB Network Is a Point to Resolve the Resistance to Targeted Therapies

The interaction between ligands and receptors is crucial in the cascade of events in HER2-positive BC. Several agents have been developed to block the biological effect caused by HER2, leading to improved prognosis for patients with HER2-positive malignance. However, they also come along with drug resistance. The mechanisms of drug resistance include loss of HER2 expression [41], insufficient HER2 downregulation [42], variation of the HER2 receptor [43], and decreased combination between drugs and HER2 [44]. In line with these situations, we summarized strategies that point to the HER2 directly. To make the description more intuitive and effortless, we collectively summarize the following strategies in Figure 3.

The SH3 domain-binding glutamic acid-rich (SH3BGR) protein family is comprised of four members: SH3BGR, SH3BGRL, SH3BGRL2, and SH3BGRL3. Research suggests that SH3BGR, SH3BGRL, and SH3BGRL2 function as tumor suppressors, while SH3BGRL3 seems to be an oncogene [45,46,47,48]. Specifically, SH3BGRL was determined to be a potential target for HER2. SH3BGRL stabilized HER2 on the cell membrane by directly binding to its motifs α1, α2 helixes, and β3 sheet, suppressing HER2 internalization and prolonging the activation of HER2 in breast cancer cells. The binding of SH3BGRL to HER2 can also cause a conformational change in the receptor, leading to maintenance of unique tyrosine phosphorylation and promoting the extension of downstream signaling, thereby promoting the proliferation and survival of breast cancer cells [49]. Moreover, it reduces the sensitivity of these cells to anti-tumor drugs, especially to HER2-targeted drugs. Effective silencing of SH3BGRL or inhibiting its downstream signals has been shown to induce apoptosis in HER2 and SH3BGRL double-positive breast cancer cells [49]. Therefore, SH3BGRL may serve as a protentional solution to drug resistance and an important therapeutic target in SH3BGRL and HER2 double-positive tumors. However, SH3BGRL has dual functions, being broadly upregulated in breast cancer but exhibiting low expression in AML progression. This dual-sided effect may be attributed to specific cell contexts in different tissues, including the HER2 expression state in breast tumors. Therefore, further clinical evidence is needed to determine the efficacy and safety of SH3BGRL in HER2-positive patients.

Interferon-γ (IFN-γ), produced by T helper 1 (Th1), was used to alleviate the resistance of HER2-targeted agents, given IFN-γ could reduce HER2 expression in the cell membrane [50]. The E3 ubiquitin ligase Cullin5 (CUL5), a tool that can break down HER2, is reduced while HER2 is overexpressed. Meanwhile, the client-protective co-chaperones cell division cycle 37 (Cdc37) and heat shock protein 90 (Hsp90) increases, protecting HER2 from the degradation of CUL5 by binding with it [51]. In above background, the experiment demonstrated the function of IFN-γ. IFN-γ increases CUL5 expression and diminishes both Cdc37 and Hsp90 from HER2 receptor, resulting in less surface HER2, further tumor senescence, and repressed tumor growth [50]. When combined with IFN-γ, the effectiveness of HER2-targeted agents will magnify, and the sensitivity will be improved. To examine the effects of IFN-γ, a clinical trial (NCT03112590) aimed at testing a combination of IFN-γ with paclitaxel, trastuzumab, and pertuzumab in HER2+ BC was initiated.

Additionally, since IFN-γ drives programmed cell death protein 1 (PD1) expression in T cells, it may make HER2-positive BC more sensitive to checkpoint therapy [52]. However, HER2-positive tumors have not been susceptible to checkpoint therapies presently. This study delivers a point that there is a crucial association between anti-HER2 CD4 Th1 immunity and HER2-mediated responses. Progressive loss of Th1 immunity was correlated with poor treatment response and prognosis in HER2-positive BC [53]. Furthermore, the interaction between tumor-antigen-specific CD4^+^T cells, cytotoxic CD8^+^T cells, and dendritic cells (DC) is tight. Conventional DC1 (cDC1) initially promotes CD4^+^T cell activation and helps the priming and infiltration of CD8^+^T cells into tumors [54]. Another study indicated that intertumoral delivery of DC combined with anti-HER2 therapy led to robust systemic antitumor immunity and complete regression in HER2-positive BC, as long as high levels of CD4+ and CD8+ T cells were present in the tumors [55].

Acetyltanshinone IIA (ATA), a chemically modified derivative of Tanshinone IIA, is a promising small molecular compound in treating HER2-positive BC. The latest study revealed that ATA increased the binding of two E3 ligases (c-Cbl and CHIP-mediated) with HER2 and led to the increased ubiquitination and degradation of HER2. Meanwhile, ATA reduced the overproducing HER2 binding partners such as HER3, IGF-1R, and MET in lapatinib-resistant cells [56]. By reducing HER2 protein expression, ATA produces a more permanent inhibitory effect on HER2’s oncogenic effect. Notably, similar to SH3BGRL as described above, ATA also exhibits a synergistic effect in inhibiting the downstream signaling pathway of HER2. In vitro and vivo experiments, ATA was more effective than lapatinib in inhibiting the growth of HER2-positive BC cells and reducing tumor growth. Moreover, ATA holds promise as a potential monotherapy agent, in contrast to most current anti-HER2 regimens which typically involve a combination of anti-HER2 drugs and other chemotherapeutics. These findings suggest that ATA may represent a new way to treat HER2-positive BC and could potentially improve the outcomes of patients who are non-responsive or resistant to current HER2 therapies. As a recent publication indicated, ATA could inhibit the tumor growth of drug-resistant lung cancer by reducing the synthesis of cell cycle-related proteins [57]. This discovery showed that ATA might have a function in a novel manner in HER2-positive BC.

Some studies found that the interplay and cross-talk between HER family and integrin cause malignancy development. In early-stage HER2-positive breast cancer patients, the upregulation of integrins will cause resistance to anti-HER2 therapies [58]. A recombinant protein named RPDC-HI has been developed to address the issue, along with its drug conjugates (doxorubicin/DOX), which target both HER2 and integrin αvβ3. RPDC-HI has been shown to effectively reduce the expressions of the HER family members, including EGFR, HER2, HER3, and HER4, as well as integrin αvβ3. Furthermore, in combination with DOX, RPDC-HI remarkably improves the tumor inhibition efficacy to 97.5% in treating HER2-positive breast cancer, compared to only 34.3% for free DOX [59]. Compared to traditional ADCs, which often have poor penetration into solid tumors, RPDC-HI exhibits optimized tumoral accumulation and diffusion, resulting in a homogeneous drug distribution throughout the tumor mass. This leads to significant reductions in tumor volume and improved efficacy against metastasis.

Usually, the HER2/HER3 heterodimers are continuously internalized from the plasma membrane and transported to early endosomes. Some of them are subsequently transported to late endosome/lysosome, where they undergo HER2/HER3 heterodimers-degradation. However, in HER2-positive breast cancer, some heterodimers are able to return to the plasma membrane through endosomal recycling, causing resistance to HER2-targeted therapies. Given the aforementioned paradigm, Anurag Mishra et al. reasoned that inhibiting the expression of Rab coupling protein (RCP), a key regulator of endosomal recycling, could help block this recycling pathway. By reducing RCP expression, ErbB family members would be diverted to the degradative pathway, leading to fewer heterodimers at the cell surface and blocking the activation of the PI3/Akt/mTOR pathway [60].

Primaquine (PQ), a small molecule inhibitor of the endosomal recycling pathway, could suppress HER2-mediated signaling. When combined with TKIs such as Lapatinib, PQ has been shown to have a synergistic effect in enhancing the activity of lysosomes and overcoming the acquired resistance [60]. Even more remarkable, the endosomal recycling pathway contributed to the recycling of other clinically relevant cell surface proteins, like integrin β1 [61], as integrin β1 could promote resistance to HER2-targeted therapies [58,62]. This result suggested that the endosomal recycling pathway has the potential to be excavated further on other resistance-relevant proteins. So, targeting the endosomal recycling pathway could have significant clinical benefits, especially for tumors with innate or acquired resistance to targeted therapies.

CMTM6 is a member of the CKLF-like MARVEL transmembrane domain-containing (CMTM) gene family and plays a crucial role in various physiological and pathological processes. CMTM6 exhibits dual biological effects in tumors. On one hand, overexpression of CMTM6 is associated with malignant molecular and clinical characteristics, and is linked to worse prognosis in several cancers, such as glioma and oral squamous cell carcinoma. On the other hand, CMTM6 has a suppressive role in colorectal and ovarian cancers.

Recent studies have suggested a link between targeted-therapy resistance and CMTM6, as CMTM6 is highly expressed in HER2-positive breast cancer and is co-localized with HER2 on the surface of breast cancer cells. In vitro and in vivo studies have shown that CMTM6 promotes survival, migration, invasion, and trastuzumab resistance of HER2-positive breast cancer cells. In patients with trastuzumab-resistant breast cancer, increased expression of CMTM6 is associated with a worse prognosis. The intrinsic mechanism behind this phenomenon is the stabilization of HER2 protein by CMTM6, which inhibits HER2 ubiquitination. HER2 ubiquitination is a process that promotes HER2 degradation [63,64]. Additionally, CMTM6 is a critical regulator of PD-L1 stability in a broad range of cancer cells, and dual inhibition of HER2 and PD-L1 successfully enhances the anti-tumor effect of anti-HER2 monotherapy in HER2-positive tumor cell [65]. This discovery indicates the potential relationship among HER2, PD-L1, and CMTM6 in breast cancer.

Bispecific antibodies (BsAb), as their name suggests, combine the functions of two monoclonal antibodies, which can bind to different targets or epitopes simultaneously, either on the same receptor or on different receptors. Therefore, it can be more powerful to target HER2 and enhance the effectiveness of existing HER2-targeted drugs to reduce resistance. It has been considered as an efficacious candidate for the treatment of HER2-positive breast cancer. We list several novel bispecific antibodies that have been discovered in the past three years.

4-1BB (CD137, tumor necrosis factor receptor superfamily 9) is a costimulatory immune receptor that can positively activate T cells and other immune cells. The activation of the costimulatory pathway leads to the potentiation of T and NK cell proliferation, cytokine production, and cytolytic activity through the stimulation of 4-1BB. Furthermore, agonism of 4-1BB can help to counteract the immunosuppressive conditions present within the tumor microenvironment. So, activation of costimulatory pathways such as 4-1BB was promising. However, clinical applications failed to demonstrate broad efficacy of 4-1BB rather than severe liver toxicity [66].

PRS-343 is a bispecific antibody-anticalin fusion protein that targets HER2 and 4-1BB, aiming to overcome these limitations and avoid unnecessary peripheral toxicity by activating 4-1BB in a tumor-specific manner [67]. Indeed, the results demonstrate that PRS-343 facilitates the 4-1BB pathway in the presence of cells that exhibit an over-expression of HER2. This implies that, to provide costimulation to T cells, PRS-343 requires more than normal physiological levels of HER2 on the target cells. At the same time, PRS-343 is the first 4-1BB bispecific to enter clinical trial development and at Phase I study (NCT03330561).

In addition, some research has excavated the potential of 4-1BB most recently. For instance, the synthetic drug comprised of EphA2 and CD137 was shown to eliminate tumors in mice, which resulted in complete responders, and benefits fight against further challenges from the tumor [68]. The anti-tumor capability of MCLA-145, a human CD137xPD-L1 BsAb, was proved to be superior to immune checkpoint inhibitor. Because MCLA-145 could activate T cells at subnanomolarand concentrations and enhances T cell priming, differentiation, and memory recall responses [69]. Given the crucial role of T cells play in regulating the effectiveness of anti-HER2 agents and the remarkable potential of 4-1BB in HER2-positive breast cancer, MCL-145 is worth further study.

M802, classified in T-cell directing/engaging bispecific (TDB), can target CD3 and HER2. It not only retains the function of T-DM1 in tumors but also recruits and activates CD3^+^T cells in clear tumor cells, as CD3^+^T cells are important in killing tumor cells [70]. Some experiments found that M802 not only retained the function of trastuzumab but also eliminated HER2-positive BC cells through recruiting CD3^+^T cells and eventually overcame the resistance of anti-HER2 agents. However, a key challenge in the clinical development of M802 is its significant clinical toxicity, including cytokine release syndrome (CRS) and neurotoxicity. Now, clinical research has initiated to detect the effect and safety of M802 in HER2-positive advanced solid tumors (NCT04501770).

The interaction between the programmed death receptor-1 (PD-1/CD279) on immune cells and its ligand, programmed death ligand-1 (PD-L1/B7-DC/CD274) on host cells, results in the suppression of the immune response, including T cell activation and proliferation, leading to escape of host cells from autoimmunity and resistance to HER2-targeted drugs [71]. Some bispecific antibodies have been discovered in this context, such as BsPD-L1xrErbB2 and anti-HER2 × PD-1 BsAb. The former enhances the efficacy of anti-HER2 monoclonal antibody therapy and increases the proportion of tumor-associated CD8+ T cells, which is beneficial to the fight against the tumor. The latter has high affinities for both HER2 and PD-1 comparable to its parent monoclonal antibodies and blocks both the HER2 signaling and the PD-1/PD-L1 interaction. It also induces ADCC and inhibits tumor proliferation, contributing to the alleviation of drug resistance [72,73].

### 3.2. Block “the Signal Processing Part”

The signal-processing parts mainly consisted of MAPK, JAK-STAT3, and PI3K/AKT/mTOR pathways. Intrinsic and acquired drug resistance to HER2-targeted therapies has been attributed to the activation of these parallel signaling pathways and the over-activation of downstream pathways, which play a critical role in developing drug resistance. From this aspect, cutting off the pathways would reduce signal processing and block the effect induced by HER2 homodimers and heterodimers (Figure 4).

The Cullin protein family, consisting of Cullin1, Cullin2, Cullin4, and Cullin7, plays a critical role in regulating cellular signaling pathways. Among these proteins, Cullin7 serves as a scaffold protein of the E3 ubiquitin ligase complex [74]. Activation of the compensation pathway IGF-1R/IRS-1 is associated with the tyrosine phosphorylation of insulin receptor substrate-1 (IRS-1). On the contrary, serine phosphorylation of IRS-1 at key sites blocks this process [49,75,76]. Cullin7 functions as a mediator, degrading serine phosphorylated IRS-1, preventing its accumulation in the cytoplasm, thereby unblocking the tyrosine phosphorylation site of IRS-1 and transmitting the signal to downstream effector molecules like PI3K. This function leads to excessive activation of downstream signals and induces resistance to Her2 inhibitors. Furthermore, Cullin7 also suppresses the expression of IGF binding protein 3 (IGFBP-3), the protein that can compete with IRS-1 for binding to IGF-1 outside the cell, which in turn can affect IGF-1R signaling. However, it is important to note that IGFBP3 modulate cell functions by both IGF-dependent mechanisms without modulation of the IGF–IGF1R signaling pathway, and they are termed IGF-1R-independent IGFBP functions [77]. Indeed, the results demonstrated that elevated levels of Cullin7 have been observed in HER2-amplified, trastuzumab-resistant breast cancer cells and tissues. Partial restoration of trastuzumab sensitivity can be achieved by reducing the expression of Cullin7 [78].

EphB4 is an erythropoietin receptor (EPO) and has proved to associate with drug resistance of cancer cells in human chronic myeloid leukemia and prostate cancer [79,80]. Overexpression of EphB4 participated in the cascade of SHP2/GAB1-MEK signals and activated c-myc, thus allowing HER2-positive BC cells to escape from lapatinib and limiting the overall drug response of lapatinib. In addition, in vivo experiments indicated that combining an EphB4 inhibitor with lapatinib is more effective in treating HER2-positive breast tumors compared to using lapatinib alone [81]. It not only provides a new target for alleviating and evaluating the resistance of HER2-targeted agents but also provides a theoretical basis for combined anti-EphB4 therapy. However, the role of EphB4 in cancer remains controversial as the Eph-ephrin interactions can both promote and inhibit tumor growth [82]. Additionally, the development of EphB4 inhibitors for clinical use faces several challenges, such as the difficulty in detecting and inhibiting the protein–protein interactions on the surface of the EphB4-ephinB2 complex with small molecules, the low selectivity of small molecular compounds targeting ATP binding sites, and the potential to block the tumor suppressor activity of EphB4 [83]. Hence, the deficiency of specificity restricts the development of EphB4 inhibitors in the clinic.

The resistance to HER2-targeted therapies in breast cancer can be addressed by inhibiting signal pathways contributing to drug resistance. As a result, the treatments of resistance elements tend to be utilized more broadly, which will induce increasingly influential HER2 inhibitor-resistant breast cancer. Cyclin-dependent kinase 7 (CDK7) is a core transcription kinase [84], which mediates the activity of multiple multi-receptor tyrosine kinase and produces resistance to HER2 targeted therapies. The inhibition of CDK7 can therefore block multiple RTK signals and prevent the reactivation of multiple kinase pathways in HER2-targeted remedy-resistant breast cancer but is not capable of silencing all the nodes of downstream intracellular signaling. Further experimentation in this area will be required.

Studies have indicated that CDK7 activity can be modulated by HER2 and RTK signaling pathways, including downstream SHP2 and PI3K/AKT nodes. Through activating CDK7, these signaling pathways regulate the phosphorylation of RNA polymerase II (RNA Pol II) C-terminal domain and confer RNA Pol II escape from promoters. Consequently, the subsequent prolonged transcripts keep maintained [85]. In addition, THZ1 is a newly discovered covalent inhibitor of the transcriptional regulatory kinase CDK7 [86]. THZ1 could strongly inhibit the growth of HER2-positive BC cells and accelerate apoptosis, even in cancer cells resistant to HER2-targeted therapies [85]. It is worth noting that another study found that dinaciclib, the cyclin-dependent kinase (CDK) inhibitor, was proved to be an effective option for myeloid cell leukemia 1 (MCL-1) blockade, as MCL-1 can mediate the resistance to HER2-targeted therapies [87]. Thus, dinaciclib made the HER2-positive BC cells susceptible to neratinib, lapatinib, and tucatinib.

G protein-coupled receptors (GPCRs) dysregulation has been linked to the development and progression of many tumors [88] due to their involvement in a wide range of physiological and pathological functions. Thus they are potential targets for therapeutic intervention [89]. However, the mechanisms underlying GPCR alteration in cancer remain largely unknown. In breast cancer cells, the expression of multiple GPCRs is altered, especially HER2 overexpression serving as a key regulator of GPCR mRNA expression [90]. This suggests that current approaches targeting multiple GPCRs may not be effective as a cancer therapeutic.

Recent studies have found that Gi/o-coupled receptors are the main group of GPCRs with altered expression in breast cancer cells. Blocking. Gi/o-GPCRs signals could increase the effectiveness of anti-HER2 therapies [91]. In HER2-positive BC, the upregulation of Gi/o-GPCRs stimulate HER2 transactivation and drive activation of effectors shared by Gi/o-GPCRs and EGFR/HER2, such as AKT and Src, promoting tumor growth and metastasis. The in vitro and in vivo experiments showed that pertussis toxin (PTx) could uncouple Gi/o and GPCRs, decrease the proliferation and migration of cells, and alleviate the malignancy development of tumors. Meanwhile, PTx combined with PI3K and Src inhibitors enhances the HER2-targeted therapy [91]. Because PI3K/AKT and Src signaling pathways, activated by Gi/o-GPCRs, play a pivotal role in the resistance of anti-HER2 drugs [92]. In summary, there may be a potential usage for PTx. However, it should be noted that PTx cannot be used as a therapeutic agent for blocking Gi/o signaling because it is a virulence factor of Bordetella pertussis and causes the respiratory disease pertussis. Therefore, further research is needed to resolve the pathogenicity of PTx or find another way to target shared signaling pathways between Gi/o-GPCRs and HER2.

Smoothened (SMO) is an oncogenic 7-transmembrane receptor that activates the glioma-associated oncogene homolog 1 (GLI1) oncogenic transcription factors. Dysregulation of the SMO-GLI1 signaling axis will cause tumorigenesis and vascular development [93]. Termed truncated GLI1 (tGLI1), an alternative splice variant of GLI1 [94], retains the known GLI1 functional domains, and promotes stemness gene expression (Nanog, SOX2, and OCT4) in breast cancer, ultimately leading to metastasis and drug resistance [95]. Consequently, SMO has become a strategic target in the treatment of several cancer types, including breast cancer. It is found that JAK2-STAT3 and SMO-GLI1/tGLI1 pathways are simultaneously activated in triple-negative breast cancer (TNBC) and HER2-enriched breast cancer [96]. Notably, the aforementioned HER2-enriched breast cancer is not equivalent to HER2-positive breast cancer, it refers to the HER2-enriched molecular subtype within the HER2-positive breast cancer. As the HER2-positive breast cancer can divided into four intrinsic molecular subtypes: luminal A, luminal B, HER2-enriched, and basal-like subtype [97]. In subsequent research, the feasibility of a combination of JAK2 inhibitors and SMO inhibitors was proved in HER2-positive BC. Currently, combinations of FDA-approved JAK2 inhibitors, including ruxolitinib (Rux) and pacritinib (Pac), and SMO inhibitors including vismodegib (Vis) and sonidegib (Son) exerted synergistic cell kill in a HER2-positive breast cancer cell lines with acquired trastuzumab resistance. Moreover, they remarkably suppressed tumor growth and lung metastasis [98].

### 3.3. “The Output Part” Is Also a Possible Anti–Resistance Strategy of HER2-Positive BC

The output part (such as cell division and migration, adhesion, differentiation, and apoptosis) is defined more extensively than the above two parts (Figure 5). There are more alternative methods for anti-resistance.

#### 3.3.1. Epigenetic

In the context of HER2, epigenetic changes, including DNA methylation and histone modification, associated with HER2-positive BC susceptibility are closely related to drug resistance [15,99]. Recent studies have shown that specific inhibitors of epigenetic factors have the potential to overcome HER2-targeted remedy resistance.

The histone demethylase, planthomeodomain finger protein 8 (PHF8), has been shown to play a significant role in breast tumorigenesis by promoting the epithelial-to-mesenchymal transition (EMT). High expression of PHF8 is associated with anchorage-independent growth [100]. Notably, in HER2-positive BC cell lines, PHF8 expression was elevated after HER2 upregulated. However, explicit rationale to explain this phenomenon has yet to be identified. Researchers speculated that the MYC-miR-22-PHF8 regulatory axis might contribute to the PHF8 upregulation. Conversely, PHF8 acted as a transcriptional coactivator to mediate HER2 expression. In other words, PHF8 and HER2 had a synergistic interplay. Notably, IL-6 plays a central role in the PHF8-related protein–protein association network and HER2 signal transduction and resistance. As a result, the PHF8-IL-6 axis has been identified as a potential mediator of resistance to anti-HER2 drugs [101]. Additionally, blocking the PHF8-TOPBP1 (DNA topoisomerase II binding protein 1) connection would trigger the vulnerability to chemotherapeutics in breast cancer [102]. In conclusion, PHF8 has been shown to play a significant role in HER2-positive BC and could serve as a potential target for combating resistance to treatment from an epigenetic perspective. Further research should be conducted to fully understand the role of PHF8 and other related mechanisms in breast cancer.

DNA methylation, a well-known epigenetic modification, has been implicated in the regulation of various crucial cancer genes [103], making it a promising epigenetic biomarker for various cancers, including BC [104]. Through genome-wide DNA methylation and transcriptomic analysis in trastuzumab-sensitive (SK) and trastuzumab-resistant (SKTR) BC cell lines, the research singled out the TGFBI gene with the highest hypermethylation-associated silencing both at the transcriptional and protein level. Overexpression of either the wild-type or mutated form of TGFBI resulted in a significant upregulation of HER1, HER2, and AKT activation, as evidenced by increased levels of phosphorylated HER1, HER2, and AKT. This was in contrast to the SKTR model, which exhibits hypermethylated TGFBI but no alteration in the overall expression of these proteins. Selectively overexpression of TGFBI in the SKTR models restored the sensitivity to trastuzumab, presenting a behavior similar to the SK model. TGFBI promoter CpG island hypermethylation is also associated with trastuzumab resistance in HER2-positive breast cancer patients. Furthermore, clinical data has shown a higher level of TGFBI methylation in patients who developed resistance after treatment. However, due to the limited number of patient cohorts, further validation in a larger and independent cohort is necessary to confirm the role of TGFBI methylation as a biomarker [105].

In normal cells, the homeostasis of histone acetylation and deacetylation are maintained by HATs and Histone deacetylases (HDACs). HDACs play a pivotal role as co-repressor of the transcriptional complex, allowing the histones to wrap the DNA more tightly and suppressing gene transcription [106]. Impeding HDAC would alleviate the resistance of anti-HER2 therapies in HER2-positive BC. The study proposed a paradigm in which HER2 signaling in HER2-overexpressing cells phosphorylates Sp1, thereby enhancing its affinity for HDAC1. The resulting Sp1-HDAC1 complex deacetylates H3K56 (histone H3 lysine 56), silences FAS and MIR146A genes and upregulates miR-146a targets such as interleukin-1 receptor-associated kinase 1 (IRAK1) and the chemokine receptor CXCR4, and ultimately promotes tumor invasion, proliferation, and survival [62]. Conversely, in the absence of HER2-signaling-phosphorylated Sp1, HDACs tend to combine with S100 and upregulate the expression of FAS and MIR146A genes, contributing to tumor suppression. In addition, the team examined a collaboration of histone deacetylase inhibitors (HDACi) and HER2-targeting in suppressing breast cancer. They found that HDACi, including trichostatin A (TSA), and suberoylanilide hydroxamic acid (SAHA/Vorinostat), could synergized remarkably with lapatinib to repress tumor growth in vitro and rodent models. A clinical trial (NCT00258349) completed in 2010 aimed to assess the effect and safety of combined vorinostat and trastuzumab. Unfortunately, the results of the trial fell short of expectations, with a response rate (RR) of 0%, a time to progression (TTP) of 1.5 months, and a mPFS of 9.3 months. To date, no clinical trials have investigated the feasibility of using lapatinib in combination with vorinostat.

The polycomb group (PcG) is a crucial epigenetic regulatory factor comprised of two multi-complexes: polycomb-repressive complex 2 (PRC2) and PRC1, with MEL-18 being a core component of PRC1. MEL-18 amplification is observed in approximately 30% to 50% of HER2-positive breast cancers and is associated with maintaining sensitivity to trastuzumab. This effect is thought to be due to the inhibition of ErbB ligand production and dimerization between ErbB receptors, mediated by epigenetic modification of ADAM family sheddases in cooperation with PcG complexes [107]. Moreover, combining ADAM10/17 inhibitors and trastuzumab may alleviate resistance in the MEL-18 loss-mediated resistance model. These findings suggest that MEL-18 amplification may serve as a novel biomarker for HER2-positive breast cancer. However, further research on MEL-18 and the ADAM family should be promoted. Some limitations include the absence of ADAM10/17 inhibitors suitable for use in combination with trastuzumab in clinical settings. The only ADAM10/17 inhibitor used in clinical trials, INCB7839 (Incell, Wilmington, DE, USA), has been suspended from further development due to a lack of apparent clinical benefit for breast cancer patients [107].

#### 3.3.2. Metabolism

Metabolic reprogramming is a key characteristic of cancer. It has been demonstrated that changes in metabolic preferences can result from pro-survival mechanisms that enable cancer cells to adjust and proliferate under stressful circumstances, including nutrient deprivation, hypoxia, or drug-induced cytotoxicity. For example, former studies indicate that genes linked to glucose depletion and glutamine metabolism were upregulated together with the development of lapatinib resistance [108,109].

HO-1 (heme oxygenase-1) is a rate-limiting enzyme that decomposes heme into biliverdin, releasing carbon monoxide and iron. Overexpression of HO-1 has been observed in numerous tumor types, including breast cancer, and is associated with a poor prognosis. Ectopic expression of HO-1 has been shown to reduce sensitivity to HER2-targeted drugs in the human HER2 overexpression cell line, potentially correlating with autophagy [110]. HO-1 could induce autophagy through signal pathways, including mTOR, ERK, JNK, and p38MAPK [111,112,113]. As a catabolic process, autophagy could activate autophagy cells and lead to acquired resistance in treatment [114]. Furthermore, some researchers have concluded that HO-1-induced autophagy plays a role in resistance to pan-HER family kinase inhibitors, indicating that HO-1 inhibitors have promising prospects and could serve as an additional treatment option for HER2-positive drug-resistant diseases [110]. However, the current difficulties lie in the limited reports of the anti-tumor activity of HO-1 selective inhibitors in vivo and the need to improve their bioavailability and effectiveness [115]. In addition, autophagy inhibitors provide another treatment option for resistance in HER2-positive breast cancer [110]. Nevertheless, in current clinical trials, autophagy inhibitors are also affected by non-target effects, limiting their usage [116].

Some researchers found that during the acquisition of resistance to HER2 inhibition, the metabolic rearrangement of breast cancer cells depends on the uptake of exogenous FA (fatty acid) rather than the de novo synthesis of FA [117,118]. Platelet glycoprotein 4 (CD36) is a membrane FA transporter that facilitates the uptake of exogenous FA and is recognized to be crucial in facilitating FAimport into cells. Additionally, CD36 is a multifunctional protein participating in various cellular signaling processes. The lapatinib-resistant cells have increased uptake of exogenous FA and metabolic plasticity. Meanwhile, in resistant cells, CD36 was found to be the most enriched protein in the Gene Ontology (GO) term “lipid metabolic process”. Subsequent experiments in vivo also showed that the absence of CD36 in the breast tissue of MMTV-neu mice could significantly reduce the occurrence of tumors, and blocking of CD36 enabled lapatinib-resistant xenograft tumors to re-sensitivity to HER2-target therapies [117]. Therefore, CD36-mediated exogenous FA uptake by cancer cells can be regarded as a primary survival mechanism of HER2-positive breast cancer. Moreover, CD36 overexpression might correlate with poor clinical outcomes in human breast cancer. From the phase III trial NeoALTTO, patients treated with trastuzumab-based therapy showed an independently associated between worse event-free survival and CD36 expression, however, this result was not fit for the patients that treated with lapatinib-based or trastuzumab-lapatinib–based therapy [119].

#### 3.3.3. Tumor Microenvironment

The tumor microenvironment closely correlates to the mechanism of drug resistance in cancer cells. Tumor stroma consists of an extracellular matrix (ECM) and different types of cells, including tumor-associated fibroblasts (TAF), immune and inflammatory cells, endothelial cells, and adipocytes [120]. Tumor cells disrupt the dynamic balance of normal tissue and change the stroma by releasing interstitial regulatory factors, such as fibroblast growth factor, thus providing cancer cells with a microenvironment to support tumor progression [121].

TAF produce and secrete high fibroblast growth factor 5 (FGF5) to activate fibroblast growth factor receptor 2 (FGFR2) in surrounding breast cancer cells. Then, FGFR2 activated HER2 through c-Src, leading to resistance of HER2 targeted therapies, considering the TAF/FGF5/FGFR2/c-Src/HER2 axis was the escape pathway of HER2 targeted therapy resistance in breast cancer. FGFR inhibitors can reverse this kind of drug resistance [122]. In addition, compared to cancer cells, TAFs were more genetically homogeneous and were less likely to develop drug resistance, making them an attractive target for cancer therapy [123]. At the same time, the experiments in vitro and in vivo models proved that trastuzumab or lapatinib, combined with FGFR2 inhibitors, was effective in overcoming resistance to targeted HER2 therapies [122].

The tumor immune environment is also crucial in regulating drug resistance in cancer. CD57^+^ natural killer (NK) cells are believed to be a potential target for improving the efficacy of HER2-specific therapeutic antibodies. In patients with HER2-positive tumors treated with trastuzumab, those with a high response rate had a higher proportion of CD57^+^ NK cells, and circulating CD57^+^ NK cells demonstrated increased trastuzumab-induced degranulation compared to their CD57^−^ counterparts [124]. On the other hand, previous studies have also shown that tumor-associated macrophages and tumor-infiltrating lymphocytes may influence patients’ response to trastuzumab [125,126].

#### 3.3.4. Cancer Stem Cells

Breast cancer stem cells (BCSCs) have been implicated in HER2-targeted drug resistance, contributing to tumor formation, infinite growth, recurrence, and metastasis [127]. Conventional drug therapies, such as chemotherapy, radiotherapy, and targeted therapy, typically only eliminate active non-stem cells, leaving CSCs with intrinsic resistance. The relatively quiescent CSC subsets in trastuzumab-resistant breast cancer represent a reservoir of tumor recurrence and a significant barrier to drug therapies [128]. Thus, targeting CSCs may offer a promising strategy to overcome resistance to HER2-targeted agents.

WEE1 is a tyrosine kinase regulatory factor at the G2/M cell cycle checkpoint. AZD1775 is an effective anticancer agent against WEE1 kinase. Some preliminary clinical studies showed that AZD1775 had a high response rate and good tolerance to toxicity [129]. A previous study confirmed that AZD1775 could induce apoptosis of trastuzumab-resistant (TRR) cells and block TRR cells in the G2/M phase. More importantly, AZD1775 had a significant inhibitory effect on CSC by preventing the expression of transmembrane mucin (MUC1) [130]. Among them, MUC1 mediated the immune escape of cancer cells and induced resistance to HER2-targeted drugs [131,132]. This finding provides new insights into the sustainable treatment of HER2-positive breast cancer by targeting CSCs. Presently, the data from clinical studies elucidate that the results of AZD1775 are encouraging, and the toxicity characteristics are acceptable in solid tumors [133]. However, there is no clinical evaluation to estimate its efficacy in treating HER2-positive breast cancer. Further research is needed to identify biomarkers for screening patients who will benefit from this treatment.

#### 3.3.5. Triggering Apoptosis induces Vulnerability of Resistant Cells

Elena Díaz-Rodríguez and colleagues established an in vitro cellular model of trastuzumab resistance to identify vulnerabilities that could help overcome resistance [134]. They made the BT474 HER2-positive breast cancer cell line secondarily resistant to the drug upon continuous exposure, a BT-RH (“Resistant to Herceptin™”) population. Genomic and proteomic analyses were conducted in both cell lines to elucidate the mechanisms underlying refractoriness. The most deregulated proteins were found to be TRAIL receptors, a member of the tumor necrosis factors (TNF) family that induces apoptosis. The subsequent in vitro experiments confirmed the increased sensitivity of BT-RH cells to TRAIL, and TRAIL-induced death-inducing signaling complex (DISC) formation was more efficient in BT-RH cells than in BT474, resulting in TRAIL killing BT-RH cells more efficiently. Although there was no information from in vivo experiments, the evidence suggests the possibility of overcoming resistance to trastuzumab by activating TRAIL receptors, which could be an advanced strategy in HER2-positive breast cancer.

Similarly, another team also put their eyes on triggering apoptosis to augment the sensitivity of trastuzumab in vivo. They previously found that in HER2-positive tumors, some bromodomain transcripts contributed to detrimental prognosis [135], as bromo and extra terminal domains (BET) family of proteins have played a role in promoting different tumors and the BET inhibitor (BETi) have been described as potential drugs to treat breast cancer [136]. Subsequently, the potential synergistic action between trastuzumab and the BETi MZ1 was explored. As a result, MZ1 was found to augment the anti-proliferative capacity of trastuzumab by DNA damage and inducing apoptosis [137]. Meanwhile, tumor volume decreased significantly after MZ1-Trastuzumab combination treatment in vivo. The MZ1-trastuzumab combination down-regulated those transcription factors relevant to poor prognosis in the HER2-positive breast cancer subtype. Taken together, inducing apoptosis is a feasible way to mitigate the resistance of targeted-HER2 agents. Notably, the apoptosis process involves many mechanisms, and these strategies deserve to be studied further.

It should be noted, however, that the mechanisms discussed above are based on basic research into the anti-resistance of HER2-targeted therapies. In order to assess their feasibility, further testing is required in preclinical and clinical trials. Unfortunately, limited data is currently available to demonstrate the effectiveness of these methods, and we summarized the existing clinical trial of aforementioned strategies (Table 3).

## 4. Conclusions

Although HER2-targeted therapies have significantly improved the survival rates of HER2-positive breast cancer patients, drug resistance remains a major challenge. The research on this issue is extensive and diverse, making it difficult to summarize and comprehend. In this review, we aim to provide a clear framework for understanding the current state of anti-HER2 therapies, highlighting the latest research hotspots and difficulties. We have organized our discussion into two parts, and we also introduce a recent development: HER2-low expression breast cancer.

Based on recent research, it appears that investigations into this issue can be divided into two main approaches: some researchers are exploring more effective synergistic combinations, while others focus on identifying novel targets or mechanisms. In this field, it is crucial not only to explore new targets and treatments horizontally but also to carry out vertical and in-depth exploration of existing research. It is worthwhile to investigate how basic research can be applied to the clinic or whether it has potential clinical applications. Some new targets and mechanisms for overcoming resistance to HER2-targeted therapies have the potential to become new biomarkers. Once they are confirmed in the clinic, they can divide HER2-positive breast cancer into different subgroups, such as SH3BGRL, TGFBI methylation, and MEl-18. This would be a significant breakthrough in the accurate and personalized treatment of HER2-positive breast cancer. Identifying biomarkers associated with drug resistance can facilitate prompt adjustments to treatment regimens, leading to better outcomes for HER2-positive breast cancer patients. However, most potential biological markers have not been studied in the patient population, thus remaining in the basic research stage. It should be noted that HER2 is currently the only valid biomarker.

The process from basic research to clinical application is lengthy because the translation of basic research into clinical applications faces significant hurdles, including identifying efficient biomarkers for usage, clarifying the toxicity and side effects, maintaining the biological activity of drugs, and improving agent selectivity. We searched for clinical trials conducted on the basic research mentioned above and summarized them in Table 3. Unfortunately, there is very little available data, which highlights one of the current difficulties in research. In addition, future research should explore how existing FDA-approved drugs can be used more effectively, including appropriate combination and dosage.

Despite the many challenges, exploring novel therapeutic strategies for HER2-positive breast cancer is of paramount importance. Such efforts can lead to the development of more effective and personalized treatments, improved patient outcomes, and a better understanding of the mechanisms underlying treatment resistance. Moreover, with the advent of precision medicine, there is a growing need to identify new targets and combination therapies that can augment the efficacy of existing treatments and surmount drug resistance. Therefore, continued investigation into the pathogenesis and treatment of HER2-positive breast cancer is indispensable to enhance patient care and ultimately diminish breast cancer-related morbidity and mortality.

## Figures and Tables

**Figure 1 cancers-15-02568-f001:**
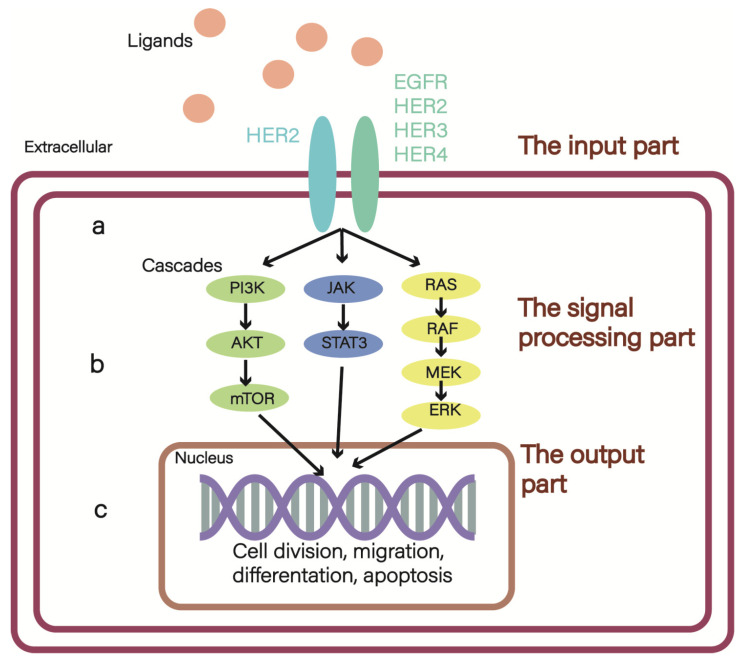
The ErbB network: (**a**) the input part, among them, the receptors have 12 kinds of combined manners, and the ErbB2-containing heterodimer is the most transforming and mitogenic receptor complex; (**b**) the signal processing part mainly includes MAPK, JAK/STAT3, and PI3K/AKT/mTOR pathways; (**c**) the output part.

**Figure 2 cancers-15-02568-f002:**
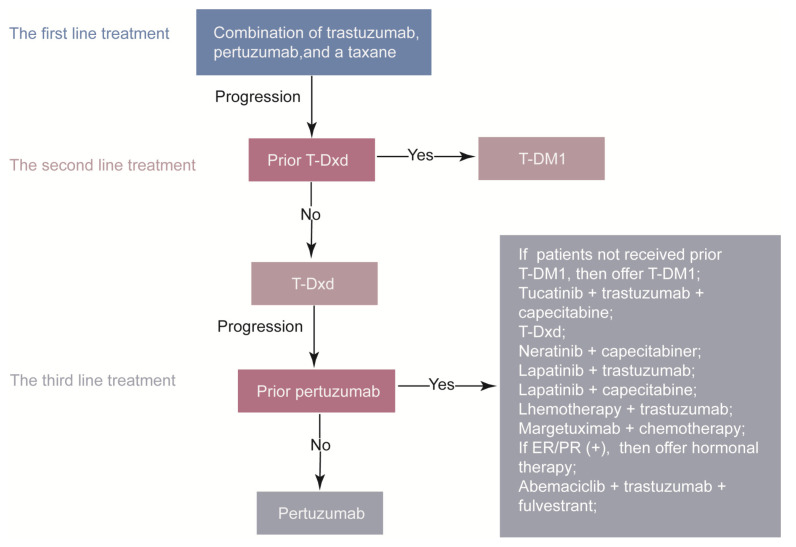
The systemic therapy for advanced HER2–positive breast cancer. ER, estrogen receptor; PR, progesterone receptor; T-DM1, trastuzumab emtansine; T-Dxd, trastuzumab deruxtecan.

**Figure 3 cancers-15-02568-f003:**
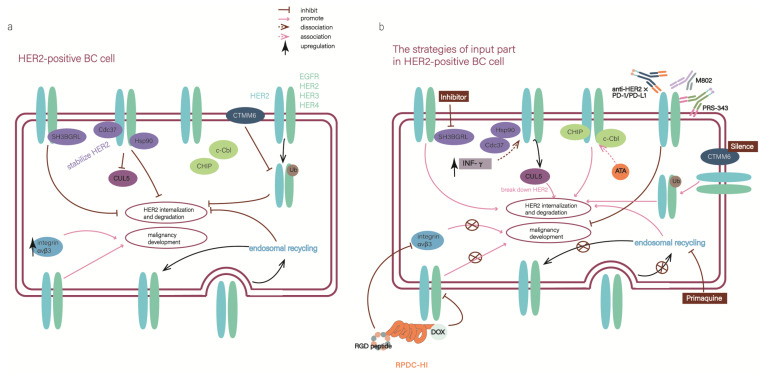
“The input part” of the ErbB network is a point to resolve the resistance to targeted therapies (**a**) The diagram illustrates the situation in HER2-positive breast cancer cells, where abnormal elements inhibit HER2 internalization and degradation, ae well as leading to the development of malignancy. (**b**) After implementing the input-part strategies, the malignancy effects of HER2 were blocked at the source.

**Figure 4 cancers-15-02568-f004:**
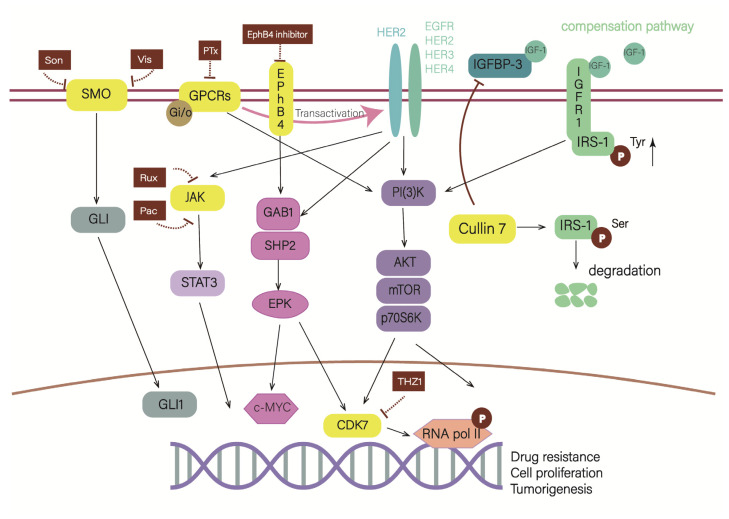
Blockade of “the signal processing part” can also alleviate drug resistance. As depicted in the schematic, Cullin 7 can promote the degradation of serine-phosphorylated IRS-1, which stimulates downstream cascades. Cullin 7 suppresses the expression of IGFBP-3, which competitively binds with IGF-1 to inhibit the activation of the IGF-1R/IRS-1 pathway. EphB4 shares a common pathway with HER2, so inhibiting EphB4 also means inhibiting the HER2 pathway. CDK7 is a core node of several HER2 pathways, indicating that blocking CDK7 can simultaneously block multiple pathways. Gi/o-GPCRs can transactivate HER2 and stimulate their common pathways, and PTx can uncouple Gi/o from GPCRs. In HER2-positive breast cancer, JAK2-STAT3 and SMO-GLI1/tGLI1 inhibitors have synergistic effect.

**Figure 5 cancers-15-02568-f005:**
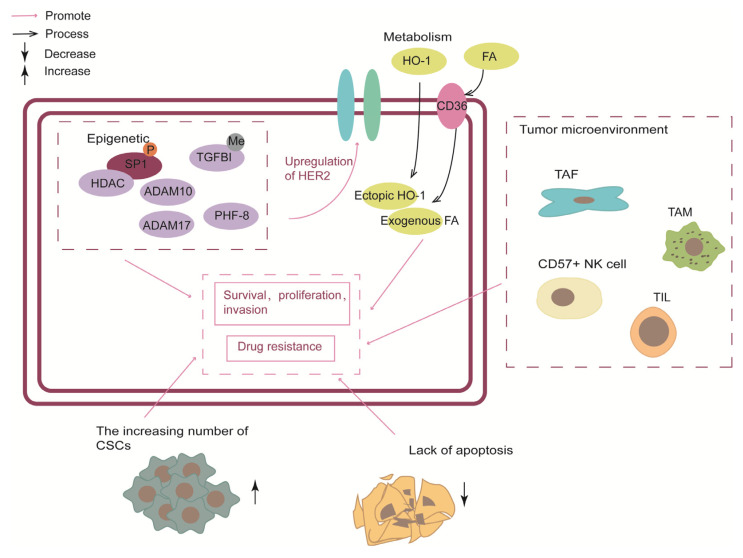
The output part was divided into five sections: epigenetic, metabolism, tumor microenvironment, cancer stem cells and triggering apoptosis. Epigenetic elements not only contribute to the malignant progression of HER2-positive breast cancer but also mediate the upregulation of HER2.CSC, cancer stem cells; TAF, tumor-associated fibroblasts; TAM, tumor-associated macrophage; TIL, tumor-infiltrating lymphocytes.

**Table 1 cancers-15-02568-t001:** Several clinical trials have enrolled both HER2-positive and HER2-low expression breast cancer patients. The table displays the trial regimens and the distinct outcomes observed between these two patient populations. BC, breast cancer; DCR, disease control rate; mPFS, median progression-free survival; HR, hormone receptor; ORR, objective response rate; TTR, time to response.

Agents	Representative Clinical Trials	Phase	Patients (*n*)	Disease	Trial Regimens	Key Results
T-Dxd	NCT02564900 [5]	Phase Ib	54	HER2 low expression BC	T-Dxd 5.4/6.4 mg/kg	ORR: 33.3% vs. 39.4%; DCR: 85.7% vs. 87.9%; TTR: 2.6 months vs. 2.7 months
		The T-Dxd has been approved by FDA to utilize in the HER2-positive BC therapy				
RC-48	NCT03052634 [6]	Phase Ib	70	HER2 low expression BC	RC48-ADC 2.0 mg/kg	ORR: 39.6%; mPFS: 5.7 months;
		Phase Ib	48	HER2-positive BC	RC48-ADC 1.5/2.0/2.5mg/kg	ORR: 22.2% vs. 42.9% vs. 36.0% mPFS: 6.2 months vs. 6.0 months vs. 6.3 months
SYD985	NCT02277717 [7]	Phase I	49	HER2 low expression BC	SYD985: 0.3 mg/kg–2.4 mg/kg (3 + 3 design)	ORR(HR+): 28%; ORR(HR−): 40%
		Phase I	50	HER2-positive BC		ORR: 33%
MCLA-128	NCT03321981 [8,9]	Phase II	40	Estrogen receptor-positive/low HER2 expression BC	MCLA-128 + endocrine therapy	DCR: 45%; common related AEs (all grades; G3–4): asthenia/fatigue (27%; 2%), diarrhea (25%; 0), nausea (21%; 0).
		Phase II	40	HER2-positive BC	MCLA-128 + trastuzumab + vinorelbine	DCR: 77%; common related AEs (all grades; G3–4): neutropenia/neutrophil count decrease (61%; 46%), diarrhea (61%; 4%), asthenia/fatigue (46%; 0), nausea (29%; 0)
SAR443216	NCT05013554 [10]	Phase I	-	Metastatic breast cancers with HER2 low expression	SAR443216	No results posted
		Phase I	-	Metastatic HER2-positive BC	SAR443216	No results posted

**Table 2 cancers-15-02568-t002:** The progress of the FDA-approved HER2 targeted agents that publish in recent 3 years. AC-KP: anthracycline-based chemotherapy, T-DM1 plus pertuzumab; AC-THP: taxane plus trastuzumab plus pertuzumab; AEs: adverse events; CBR: clinical benefit rate; CNS: central neural system; CRTs: clinically relevant toxicities; IDFS: invasive disease-free survival; MBC: metastatic breast cancer; mOS: median overall survival; mPFS: median progression-free survival; NR: non-responder; ORR: objective response rate; TH: paclitaxel plus trastuzumab; tpCR: total pathologic complete response.

NCT Number	Phase	Trial Regimens	Enrolled Population	Actual Patients (*n*)	Status	Key Results of Effect	Key Results of Toxicity
NCT01912963 [29]	Phase II	Trastuzumab + pertuzumab + eribulin in Cohort A: (no prior pertuzumab) vs. Cohort B: (prior pertuzumab)	Metastatic, unresectable locally advanced, or locally recurrent HER2-positive breast cancer	32	Terminated	ORR: 26.3% vs. 0%; mOS: 19.9 vs. 28.4 months, mPFS: 7.5 vs. 3.4 months	Grade 3 AEs (neutropenia, dehydration, fatigue, diarrhea anemia, peripheral sensory neuropathy) were more common in Cohort B
NCT02586025 [30]	Phase III	Trastuzumab + pertuzumab + docetaxel vs. trastuzumab + placebo + docetaxel	Early stage or locally advanced HER2-positive breast cancer	329	Active, not recruiting	tpCR: 39.3% vs. 21.8%; ORR: 88.6% vs. 78.2%	Diarrhea: 38.5% vs. 16.4%; grade ≥ 3 AEs (Neutropenia, Leukopenia): 48.6% vs. 41.8%
NCT02536339 [31]	Phase II	Trastuzumab (high dose: 6 mg/kg weekly) + pertuzumab	HER2-positive MBC with CNS metastases and CNS progression	39	Completed	ORR: 11%; CBR at 4 months was 68%; CBR at 6 months was 51%	Grade ≥ 1 AEs (diarrhea, fatigue, nausea): 97%; No grade 5 AEs
NCT02614794 [23]	Phase II	Tucatinib + capecitabine + trastuzumab vs. trastuzumab + capecitabine + placebo	Pretreated unresectable locally advanced or metastatic HER2-positive breast cancer	612	Active, not recruiting	mOS: 21.9 vs. 17.4 months, mPFS: 7.8 vs. 5.6 months	Diarrhea was the most common AEs in both two group: grade 1: 43.3% vs. 32.0%; grade 2: 24.8% vs. 12.7%; ≥grade 3: 12.9% and 8.6%
NCT00684983 [32]	Phase II	Arm A: Apecitabine + lapatinib + cituxumumab vs. Arm B: apecitabine + lapatinib	HER2-positive advanced breast cancer	68	Completed	Apecitabine + lapatinib + cituxumumab did not improve PFS or OS comparired with apecitabine + lapatinib	Grade 3 AEs: alanine aminotransferase increased (16% vs. 3%), aspartate minotransferase increased (16% vs. 3%), diarrhea (26% vs. 11%); Hand-and-foot syndrome/reaction (5% vs. 25%); Grade 4 AEs: 5.3% vs. 0%
NCT01808573 [33]	Phase III	Neratinib + capecitabine vs. lapatinib + capecitabine	HER2-positive metastatic breast cancer	621	Completed	PFS: 8.8 months vs. 6.6 months; OS: 24.0 months vs. 22.0 months	Serious AEs (diarrhea, nausea, palmar-plantar erythrodysesthesia syndrome, and vomiting): 34% vs. 30%
NCT01966471 [34]	Phase III	AC-THP vs. AC-KP	Early HER2-positive breast cancer	1846	Completed	IDFS in node-positive subpopulatio: 82% vs. 80%; IDFS in overall population: 88% vs. 86%;	Grade ≥ 3 AEs (Neutropenia mainly): 55.4% vs. 51.8%; serious AEs: 23.3% vs. 21.4%
NCT01702558 [35]	Phase II	T-DM1 + capecitabine vs. T-DM1	Metastatic HER2-positive breast cancer with previously treated	182	Terminated	ORR: 44% vs. 36%	AEs: 95% vs. 88% grade 3–4 AEs (thrombocytopenia, AST increased, γ-glutamyltransferase increased): 44% vs. 41%, no grade 5 AEs were reported
NCT01702571 [36]	Phase III	The tolerability and efficacy of T-DM1 in brain metastatic HER2-positive breast cancer	Brain metastatic HER2-positive breast cancer	2185	Completed	With baseline BM: mOS: 18.9 months, mPFS: 5.5 months; serious AEs: 28.4% Without baseline BM: mOS: 30 months, mPFS: 7.7 months; serious AEs: 19.6%	With baseline BM: serious AEs: 28.4%, headache and vomiting were common Without baseline BM: serious AEs: 19.6%, nervous system AEs were common

**Table 3 cancers-15-02568-t003:** Basic research strategies that have entered the stage of clinical trials. Utomilumab: also known as 4-1BB Agonist Monoclonal Antibody, PF-05082566, PF-05082566, PF-2566; PRS-343: also known as HER2/4-1BB Bispecific.

Agents	Targets	Representative Clinical Trials	Status	Trial Regimens	Phase	Disease Condition	Patients (*n*)
**IFN-γ**		NCT03112590	Active	IFN-γ + paclitaxel + trastuzumab + pertuzumab	Phase II	HER2-positive breast cancer (male, female, post therapy surgery)	51
**Utomilumab**	4-1BB	NCT03364348	Completed	trastuzumab + utomilumab (20 mg or 100 mg/kg) vs. T-DM1 + utomilumab (20 mg or 100 mg/kg)	Phase I	HER2-positive breast carcinoma	18
**PRS-343**	HER2, 4-1BB	NCT03650348	Unknown	PRS-343 + atezolizumab	Phase I	HER2-positive solid tumor (breast cancer, gastric cancer, bladder cancer)	45
		NCT03330561	Completed	PRS-343	Phase I	HER2-positive solid tumor (breast cancer, gastric cancer, bladder cancer)	85
**M802**	HER2, CD3	NCT04501770	Not yet recruiting	M802	Phase I	HER2-positive solid tumor	32

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
