# Peer review of "Preclinical and Basic Research Strategies for Overcoming Resistance to Targeted Therapies in HER2-Positive Breast Cancer"

_cancers, 2023, doi:10.3390/cancers15092568_

Round 1

Reviewer 1 Report

In this review, Yi Cao et al. have extensively investigated the field of resistance to anti-HER2 targeted therapies especially focusing on the potential strategies to overcome them. Even though this is a good approach to the topic, it has been very difficult to me following the paper mainly due to the poor English usage. This must be extensively edited before further progressing.

In addition, figures might also be improved since they are quite poor. In fact, for example figure 2 is very small and difficult to read in its present form. Besides, all the figure legends should be homogeneous and include a better and more accurate and clear description of what is shown on each of them.

I do not understand the three blocks in which authors separate the strategies to overcome resistance. In fact, to me, some of the strategies described should be changed and included in a different block. Thus, for example SH3BGRL or ATA should be better included in the “signalling” category instead of the in the input part. I think that these three blocks should be better explained and justified in the text and in the figure legends. In line with that, it could also be a good idea to summarize the results for the third block in one or two extra figures.

I also miss a table including the different clinical trials, drugs and results in the HER2 positive breast cancer context similar to the one shown for the HER2-low (table 1).

Besides, It could probably be included a global approach discussing on the present  known mechanisms of resistance to all the anti-HER2 therapies and then dissecting the potential strategies to overcome them.

Finally, there are several important basic concept errors that must be corrected, such as describing SH3BGRL as an ErbB2 ligand or speaking about lapatinib as a HER2 TKI.

Author Response

Dear Reviewer,

We appreciate your time and effort in carefully reviewing our manuscript and providing valuable feedback and suggestions. We have carefully considered your comments and make necessary revisions to the manuscript. If you have any additional suggestions or feedback, we would be more than happy to hear them and incorporate them into our work.

Here are our responses to the comments:

  1. The poor English usage must be extensively edited before further progressing

 We are very sorry about that the poor English usage not give you a good reading experience. In this version of my manuscript, we rewrite it again and hope that you and any other readers have no problem in reding it.

  1. Figures should be improved since they are quite poor.

We have improved distinguishability of all figures, maybe this time, it can be identified clearly.

  1. Why is basic research divided into three parts, and why SH3BGRL and ATA are put into "input"?

Due to the complex and intertwined mechanisms of drug resistance and the extensive research on its underlying mechanisms, we began to scrutinize the information we had collected when we learned that the HER2 pathway could be divided into three distinct parts. It appeared that the basic research on overcoming drug resistance could also be summarized within these three levels. Therefore, we plan to use this hierarchical approach to explain the research, which will enable readers to have a general framework of understanding on basic research related to drug resistance.

We included SH3BGRL and ATA in the "input" category for the following reasons:

Firstly, SH3BGRL directly binds to the α1 and α2 helices, as well as the β3 sheet of HER2 motifs, thereby stabilizing HER2 on the cell membrane. This reduces the internalization of HER2 and prolongs its activation in breast cancer cells.

Secondly, ATA increases the binding of two E3 ligases to HER2, which in turn enhances the ubiquitination and degradation of HER2.

Finally, all of the aforementioned mechanisms affect the structural fluctuations of the HER2 receptor, ultimately leading to downstream effects. Therefore, we categorized them under "input".

  1. The third block should summarize the results in one or two extra figures.

Thanks for your suggestion, and we have added the Figure 5 to summarize the third block.

  1. Supported a table including the different clinical trials, drugs and results in the HER2 positive breast cancer context.

  We have added the necessary content in Table 2.

  1. It could probably be included a global approach discussing on the present known mechanisms of resistance to all the anti-HER2 therapies and then dissecting the potential strategies to overcome them.

If it weren't for your suggestion, we would have overlooked this point. To avoid making the article excessively lengthy, we summarized it in Figure 2.

  1. There are several important basic concept errors that must be corrected, such as describing SH3BGRL as an ErbB2 ligand or speaking about lapatinib as a HER2 TKI.

Thank you for your careful attention, which helped me identify the errors that needed to be corrected. After reviewing the original text, we found that it was incorrect to refer to SH3BGRL as a receptor of ErbB2. we have made the necessary revisions to the original text to correct this mistake. With regard to lapatinib, we have modified it in the text to refer to it as an EGFR/HER2 TKI.

Once again, thank you for your hard work and patience. We have great respect for your expertise and dedication. Please accept my apologies and gratitude. We hope that the revised version will meet your expectations and look forward to feedback from you soon.

Best regards,

Yi Cao

Reviewer 2 Report

This review article by Yi Cao et al, provides the current state of preclinical and basic research strategies for overcoming resistance to targeted therapy in HER2-positive breast cancer. The authors present a comprehensive analysis of the current understanding of the mechanisms of resistance and the potential approaches for developing new therapies.

Major Comments:

1)  The review article could benefit from more specific examples and case studies that illustrate the effectiveness of different preclinical and basic research strategies. This could help readers better understand the practical implications of the research and how it can be applied in clinical settings.

2) The review authors should consider providing more detail on the challenges and limitations of preclinical and basic research strategies for overcoming resistance to targeted therapy in HER2-positive breast cancer. This could include potential side effects and toxicities associated with different therapies, as well as the practical challenges of translating preclinical research into clinical practice.

3) The review article could benefit from a more detailed discussion of the implications of the research for personalized medicine and precision oncology. This could include how preclinical and basic research strategies can be used to identify patient-specific biomarkers that can inform treatment decisions and improve patient outcomes.

Minor Comments:

1)  Table 1 and figure 2 are blurred. Please provide clear table 1 and figure 2.

2)  English language editing should be done.

3)    Add latest references

Author Response

Dear Reviewer,

We appreciate your time and effort in carefully reviewing our manuscript and providing valuable feedback and suggestions. We have carefully considered your comments and make necessary revisions to the manuscript. If you have any additional suggestions or feedback, we would be more than happy to hear them and incorporate them into our work.

Here are our responses to the major comments:

  1. The review article could benefit from more specific examples and case studies that illustrate the effectiveness of different preclinical and basic research strategies. This could help readers better understand the practical implications of the research and how it can be applied in clinical settings.

Thank you very much for your suggestion. It is essential for our review. Regarding the comments you raised, we consulted specific cases and made supplements accordingly. It is worth noting that we also pointed out that there are few studies on clinical trials of basic research. We have summarized these studies in Table 3.

  1. The review authors should consider providing more detail on the challenges and limitations of preclinical and basic research strategies for overcoming resistance to targeted therapy in HER2-positive breast cancer. This could include potential side effects and toxicities associated with different therapies, as well as the practical challenges of translating preclinical research into clinical practice.

In response to this comment, we conducted a new search to identify every mechanism mentioned in the basic research, and we added the corresponding limitations and challenges to the end of the relevant subsections. Similarly, in the first part of the manuscript, we added "Key results of toxicity" to Table 2 to prevent the article from being too long. Additionally, the tabular format is more convenient for reading and comparison.

  1. The review article could benefit from a more detailed discussion of the implications of the research for personalized medicine and precision oncology. This could include how preclinical and basic research strategies can be used to identify patient-specific biomarkers that can inform treatment decisions and improve patient outcomes.

After some consideration, we realized that this is exactly what the conclusion section of the manuscript was missing. Therefore, we added it to the conclusion section to improve the conclusion part more effectively.

Here are our responses to the minor comments:

  1. Table 1 and figure 2 are blurred. Please provide clear table 1 and figure

We have now updated all the pictures in our manuscript to provide clearer versions. We apologize for any inconvenience caused by the previous blurred images, and hope that the new images will better facilitate the understanding of our manuscript.

  1. English language editing should be done.

We have taken this feedback seriously and have carefully reviewed and corrected the English language throughout the manuscript. We have also had the manuscript reviewed by a professional editor to ensure that the language is clear and concise.

  1. Add latest references

Based on the comments, we have updated our manuscript by adding several latest references related to our research topic.

Thank you once again for your helpful comments, which has been instrumental in improving our manuscript. We hope that our revised version will meet your expectations.

Best regards,

Yi Cao

Reviewer 3 Report

In the present article titled “Preclinical and Basic Research Strategies for Overcoming Resistance to Targeted Therapy in HER2-Positive Breast Cancer” authors reviewed the different Strategies for Overcoming Resistance to Targeted Therapy in HER2-Positive Breast Cancer. Following are some points need to be addressed.

 1. In Table 1, authors covered the clinic trials of targeted therapies in HER2 low expression breast cancer. Similarly authors should generate another Table explaining the information about the basic research strategies implemented. 

2. Add a paragraph stating the future research implications on the target therapy in HER2-positive breast cancer. 

3. Improve the conclusion part more effectively. 

4. Correct the references and make them in a constant format.

Author Response

Dear Reviewer,

Thank you for taking the time to review our manuscript and providing valuable feedback. We appreciate your comments and suggestions, which have helped us to improve the quality of our work.

Here is our response to the comments:

  1. In Table 1, authors covered the clinic trials of targeted therapies in HER2 low expression breast cancer. Similarly, authors should generate another Table explaining the information about the basic research strategies implemented. 

Thank you for your insightful feedback. We apologize for not including another table to summarize the basic research strategies of HER2-low expression BC. However, we have made changes to Table 1 to ensure that it fits smoothly in the context. The initial version of Table 1 only showed clinical trials of targeted therapies in HER2-low expression breast cancer. The revised version of Table 1 now includes clinical trials of targeted therapies in both HER2-positive and HER2-low expression breast cancer patients.

  1. Add a paragraph stating the future research implications on the target therapy in HER2-positive breast cancer. 

Thank you for your valuable suggestion. We have added your suggestion to the conclusion section, specifically in the last paragraph.

  1. Improve the conclusion part more effectively. 

Upon reviewing our manuscript, we realized that your suggestion was very insightful. Indeed, the conclusion was too simplistic. Therefore, we have added the issues you raised in the previous suggestion and added future research directions for targeting HER2, as well as the challenges and limitations that exist.

  1. Correct the references and make them in a constant format.

We have checked the citation format for all references and we appreciate your attention to detail. We found inconsistencies in the citation format and have made the necessary corrections in this version of the manuscript.

Once again, we appreciate your thoughtful feedback and hope that the revised manuscript meets your expectations.

Best regards,

Yi Cao

Round 2

Reviewer 1 Report

This review investigates the field of resistance to anti-HER2 targeted therapies especially focusing on resistances to such treatment and the potential strategies to overcome them.

In the present version of the manuscript, authors have partially corrected the English usage along the manuscript, but it still needs to be thoroughly reviewed. For example, to begin with, title should be changed to a better one, or at least to ”Preclinical and Basic Research Strategies for Overcoming Resistance to Targeted THERAPIES in HER2-positive Breast Cancer”. Or for example, the beginning of the introduction should read: HER2 (ALSO known as Neu or ErbB2). And like this, many more changes should be incorporated.

The addition of new figures has improved the manuscript. Nonetheless, and in the case of figure 2, it is still impossible to read since it is larger than the document. 

On the other hand, tables are of low quality and impossible to read, mainly table 2.

As explained in the first review, appropriate description of the figures and tables should be included in figure legends.

As I previously suggested the structure of the paper should include a description of all the known mechanisms of resistance to the different anti-HER2 therapies including mAbs, TKIs and ADCs, and then discuss the proposed strategies to overcome them.

There are still concept errors along the manuscript that should be eliminated.

Author Response

Dear Reviewer,

Thank you for taking the time to review our manuscript and provide valuable feedback. We appreciate your high-quality comments, which have helped us to improve the quality of the manuscript. We have carefully considered your suggestions and made the necessary revisions.

Here are our responses to the comments:

  1. The English usage still needs to be thoroughly reviewed

Thank you for pointing out the problem with our English usage. We have thoroughly reviewed and improved the language throughout the manuscript.

  1. Figure 2 is still impossible to read since it is larger than the document.

We have relocated Figure 2 and resized it to ensure it can be fully displayed in the manuscript.

  1. Tables are of low quality and impossible to read, mainly table 2.

We agree that Table 2 was of low quality due to it being shrunk too small. We have replaced Table 2 with an editable table.

  1. Appropriate descriptions of the figures and tables should be included in figure legends.

Thank you for your suggestion, we have included more detailed and necessary annotations in the figure legends.

  1. The structure of the paper should include a description of all the known mechanisms of resistance to the different anti-HER2 therapies including mAbs, TKIs and ADCs, and then discuss the proposed strategies to overcome them.

Combining your suggestion and revising our manuscript again, it is really necessary to add the known mechanisms of HER2-targeted therapies. We have included it in Part 1.

  1. There are still concept errors along the manuscript that should be eliminated.

Thank you for highlighting these errors. We have corrected the errors, including the mislabeling of Lapatinib as a HER2 TKI instead of a HER2/EGFR TKI and clarifying the description of IGFBP-3. We have also italicized the HER2 gene to differentiate from the protein.

We appreciate your patient guidance and suggestions, which have helped us to identify the shortcomings of the manuscript. We hope that the revised manuscript meets your expectations and we look forward to receiving your feedback.

Thank you again for your valuable time and efforts, and we wish you all the best.

Best regards,

Yi Cao

Reviewer 2 Report

Thank for your patiently incorporating the changes.

Author Response

Dear Reviewer:

Thank you for your meticulous and patient suggestions before, which have been extremely valuable to our manuscript. Wish you a happy life!

Best regards,

Yi Cao